# Clinicians' communication with patients receiving a MCI diagnosis: The ABIDE project

Leonie N. C. Visser [1,2]☯*, Ingrid S. van Maurik [2,3]☯, Femke H. Bouwman[2], Salka Staekenborg[4], Ralph Vreeswijk[5], Liesbeth Hempenius[6], Marlijn H. de Beer[7], Gerwin Roks[8], Leo Boelaarts[9], Mariska Kleijer[10], Wiesje M. van der Flier [2,3], Ellen M. A. Smets[1]

**1** Department of Medical Psychology, Amsterdam Public Health research Institute, Amsterdam UMC, University of Amsterdam, Amsterdam, the Netherlands, **2** Alzheimer Center Amsterdam, Department of Neurology, Amsterdam Neuroscience, Vrije Universiteit Amsterdam, Amsterdam UMC, Amsterdam, The Netherlands, **3** Department of Epidemiology and Biostatistics, Vrije Universiteit Amsterdam, Amsterdam UMC, Amsterdam, The Netherlands, **4** Department of Neurology, Tergooi Ziekenhuis, Blaricum, The Netherlands, **5** Department of Clinical Geriatrics, Spaarne Gasthuis, Haarlem, The Netherlands, **6** Geriatric Center, Medical Center Leeuwarden, Leeuwarden, The Netherlands, **7** Department of Neurology, Reinier de Graaf Gasthuis, Delft, The Netherlands, **8** Department of Neurology, Elisabeth-TweeSteden Ziekenhuis, Tilburg, The Netherlands, **9** Geriatric Department, NoordWest Ziekenhuis Groep, Alkmaar, The Netherlands, **10** Department of Neurology, LangeLand Ziekenhuis, Zoetermeer, The Netherlands

☯ These authors contributed equally to this work.
* n.c.visser@amsterdamumc.nl

**Data Availability Statement:** All relevant data are within the manuscript and Supporting Information files.

# Abstract

## Background

We aimed to explore clinicians' communication, including the discussion of diagnosis, cause, prognosis and care planning, in routine post-diagnostic testing consultations with patients with Mild Cognitive Impairment (MCI).

## Methods

Thematic content analysis was used to analyze audiotaped consultations in which 10 clinicians (eight neurologists and two geriatricians) from 7 memory clinics, disclosed diagnostic information to 13 MCI patients and their care partners. We assessed clinician-patient communication regarding diagnostic label, cause, prognosis and care planning to identify core findings.

## Results

Core findings were: clinicians 1) differed in how they informed about the MCI label; 2) tentatively addressed cause of symptoms; 3) (implicitly) steered against further biomarker testing; 4) rarely informed about the patient's risk of developing dementia; 5) often informed about the expected course of symptoms emphasizing potential symptom stabilization and/or improvement, and; 6) did not engage in a conversation on long-term (care) planning.

**Funding:** This study is funded by ZonMW-Memorabel (ABIDE project; project lead WMvdF; project no 733050201), in the context of the Dutch 'Deltaplan Dementie', and the EU Joint Programme Neurodegenerative Disease Research (ADDITION project; to WMvdF and ISvM; grant no. 733051083). The funders had no role in study design, data collection and analysis, decision to publish, or preparation of the manuscript.

**Competing interests:** Leonie N.C. Visser, Ingrid S. van Maurik, Femke H. Bouwman, Salka Staekenborg, Ralph Vreeswijk, Liesbeth Hempenius, Marlijn H. de Beer, Gerwin Roks, Leo Boelaarts, Mariska Kleijer, and Ellen M.A. Smets have declared that no competing interests exist. Prof. dr. Wiesje M. van der Flier has the following competing interests: she performs contracted research for Biogen MA Inc. Her research programs have been funded by ZonMW, Health Holland, Pasman stichting, NWO, EU-FP7, EU-JPND, Alzheimer Nederland, Cardiovasculair Onderzoek Nederland, stichting Dioraphte, Gieskes-Strijbis fonds, Boehringer Ingelheim, Piramal Neuroimaging, Roche BV, Janssen Stellar, and Combinostics. All funding is paid to the institution.

**Abbreviations:** AAN, American Academy of Neurology; ABIDE, Alzheimer Biomarkers in Daily Practice; AD, Alzheimer's Disease; CSF, Cerebrospinal fluid; CT, Computer tomography; MCI, Mild cognitive impairment; MMSE, Mini-mental state examination; MOCA, Montreal Cognitive Assessment; MRI, Magnetic resonance imaging; PET, Positron Emission Tomography.

## Discussion

Clinicians' information provision about the underlying cause, prognosis and implications for long-term (care) planning in MCI could be more specific. Since most patients and care partners have a strong need to understand the patient's symptoms, and for information on the prognosis and implications for the future, clinicians' current approach may not match with those needs.

## Introduction

In recent decades, improvement in the diagnostic work-up and understanding of the slow development of Alzheimer's disease (AD) caused a shift to an earlier diagnosis [1]. The diagnostic label of mild cognitive impairment (MCI) has been introduced to describe a syndrome in which cognitive impairment is objectified, but there is no dementia, and independence in functional abilities is intact [2]. The recently published practice guideline by the American Academy of Neurology (AAN) states that an accurate diagnosis of MCI is important for patients and care partners to understand the cause of their complaints, to discuss the prognostic possibilities (i.e. development of symptoms and/or risk of dementia) and to arrange long-term (care) planning accordingly [3].

Nonetheless, the MCI label remains challenging for clinicians. First, due to much heterogeneity, this diagnostic label does not imply one cause. MCI may be the first cognitive expression of AD, but may also be secondary to other neurological or psychiatric disorders [4]. Diagnostic tests, e.g. CSF AD biomarkers, can help to determine the origin of MCI [5]. However, this raises a second challenge; as MCI patients do not (yet) have dementia, the label of MCI entails a prognosis rather than a diagnosis. In general, prognostic information is difficult for clinicians to convey, and for patients and care partners to understand [6, 7]. It may even be more difficult in dementia, as no fixed events define its onset, which makes it particularly hard to identify transition points for individual patients [4].

Recently, recommendations were published on how to deliver AD biomarker results and subsequent prognostic information in MCI patients [8]. They state that a positive result, indicative for underlying AD pathology, should elicit further monitoring and conversations about future (care) planning. In the case of a negative result, they advise to communicate the uncertainty that is inherent to a negative result, i.e. that patients remain at increased risk for dementia. Not all clinicians make use of AD biomarkers however, resulting in practice variation in the diagnostic work-up [9, 10].

Despite aforementioned challenges, accurate communication about the MCI label, underlying cause of symptoms, prognosis and long-term (care) planning are recommended [3]. Currently, empirical observational data on whether and how clinicians deal with these challenges when informing patients and their care partners about these issues in clinical practice are lacking. In this qualitative study, our objective was to explore clinicians' diagnostic communication, including the discussion of cause, prognosis and care planning, in post-diagnostic testing consultations with MCI patients.

## Methods

### Design and study context

We used a qualitative approach to analyze audiotaped clinician-patient consultations in which clinicians disclosed the diagnostic label MCI to patients and their accompanying care partners.

This study was part of the Alzheimer's Biomarkers in Daily Practice (ABIDE) project [11]. ABIDE encompasses an observational study of audiotaped consultations during the routine diagnostic work-up at Dutch memory clinics [12]. Written informed consent was obtained from clinicians, patients and care partners. The board of the Medical Ethics Committee of the Amsterdam UMC reviewed and approved this study.

## Sample, procedure and questionnaires

All clinicians involved in diagnostic consultations at seven Dutch memory clinics were eligible and invited to participate (self-selection). The newly referred patients of consenting clinicians (in the Netherlands patients are referred to the memory clinic by another medical doctor, i.e. no self-referral) and accompanying care partners were invited to participate. Next, audio-recordings were made of the clinician-patient consultations when patients were seen as part of normal clinical care at the memory clinic. Since our aim was to observe and describe naturalistic clinician-patient communication during the routine diagnostic work up for dementia in a broad range of memory clinics, no efforts were made to intervene or standardize the diagnostic work up (for a detailed description see Visser et al., [12], and the results section for details on diagnostic tests that patients underwent). For the current study we selected fully audio-recorded post-diagnostic testing consultations with MCI patients (N = 13), as based on their retrospectively obtained medical records (see Fig 1). These were patients labeled explicitly with MCI (N = 11), prodromal AD (N = 1) or objective cognitive disorder(s) without meeting the criteria for dementia (N = 1). We also retrieved patient characteristics, MMSE score and which diagnostic tests were performed from medical records. Clinicians (N = 10) involved in these consultations completed a questionnaire assessing their age, gender, specialty, and level of experience.

## Analysis

We used descriptive statistics to report characteristics of clinicians, patients and care partners. Two of the authors (LNCV and ISvM, with a background in psychology) listened to all consultations twice and transcribed clinician-patient interactions related to communication about the MCI label, cause, prognostic information and future (care) planning. Qualitative content analysis was used to systematically organize the data [13]. First, an initial coding scheme was developed, based on previous research [14, 15], literature/guidelines on MCI [3, 5, 8], and literature on risk communication/provision of prognostic information [7], to deductively categorize the observed communication. The study-specific coding scheme (S1 Appendix) assessed clinician-patient communication regarding:

- diagnostic label, i.e., whether clinicians used the term MCI or Mild Cognitive Impairment, and addressed each of three MCI criteria [2, 5]; 1) observed cognitive impairment, 2) no interference with daily living, and 3) not demented;

- cause, i.e. whether and which diagnostic test results were communicated, and whether and which (potential) etiology was discussed by the clinician;

- prognosis, i.e., whether and how potential symptom development and the risk of developing dementia were discussed;

- next steps in (care) planning, i.e., whether and which next steps were addressed by the clinician.

All transcripts were independently coded by LNCV and ISvM. Codings were then compared and discussed until consensus was reached. The final codes combined with the relevant

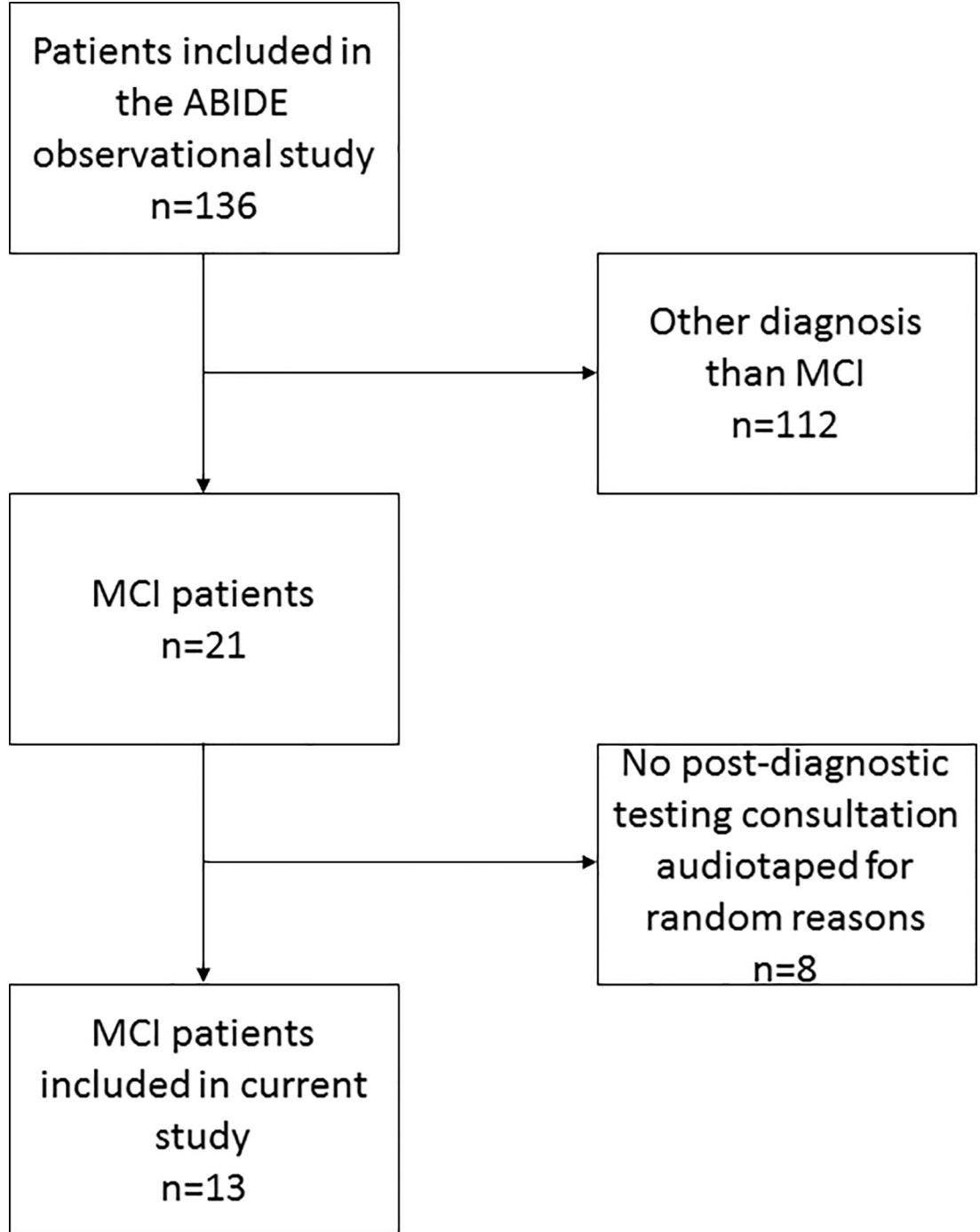

**Fig 1. Flow diagram of patient inclusion.** *Note.* Of the total sample of patients participating in the ABIDE observational study [12], 21 (15%) were diagnosed with mild cognitive impairment (MCI), based on their retrospectively obtained medical record data. Post-diagnostic testing audio recordings were available from 13/21 MCI patients, who were seen by ten clinicians in seven Dutch memory clinics.

transcribed segments of the audio-recordings were then inductively categorized further by LNCV, ISvM and EMAS, identifying core findings emerging from the data. Selected representative quotes were translated from Dutch to English by the VU university translating office.

**Table 1. Sample characteristics.**

| Patient ID | Age | Gender | MMSE | Duration of consultation (min) | Care partner (Female/Male) | Clinician |
|---|---|---|---|---|---|---|
| 1 | 70 | Male | 28 | 17 | Spouse/partner (F) | Neurologist 1 |
| 2 | 65 | Female | 26 | 17 | Spouse/partner (M) | Neurologist 2 |
| 3 | 81 | Female | 21 | 19 | Spouse/partner (M) | Neurologist 3 |
| 4 | 63 | Female | 22 | 16 | Son (in law) (M) | Neurologist 4 |
| 5 | 88 | Female | 17 | 28 | Daughter (in law) (F) | Neurologist 4 |
| 6 | 91 | Female | 27 | 14 | Daughter (in law) (F) | Geriatrician 1 |
| 7 | 73 | Male | MOCA 22/30 | 16 | Spouse/partner (F) | Neurologist 5 |
| 8 | 73 | Male | 28 | 14 | Spouse/partner (F) | Neurologist 5 |
| 9 | 73 | Male | 29 | 9 | Spouse/partner (F) | Neurologist 6 |
| 10 | 73 | Male | 29 | 16 | Spouse/partner (F) | Neurologist 7 |
| 11 | 64 | Male | 29 | 38 | Spouse/partner (F) | Geriatrician 2 |
| 12 | 79 | Female | 29 | 13 | Son/Daughter (in law) (F) | Geriatrician 2 |
| 13 | 61 | Male | 26 | 27 | Spouse/partner (F) | Neurologist 8 |

*Notes*. Each row represents a patient and the patient ID represents the same patient across all tables. MMSE = Mini-mental State Examination, MOCA = Montreal Cognitive Assessment.

## Results

Table 1 presents characteristics of the 13 MCI patients. They were all accompanied by a care partner, and seen by eight neurologists and two geriatricians. Mean age of patients was 73±9, 6/13 were female and their mini mental state examination (MMSE) score was 26±4 (M±SD). Mean age of clinicians was 49±9 (M±SD), 4/11 were female, they had 10±8 (M±SD) years of experience working in a memory clinic and saw on average 18±8 (M±SD) new patients per month. Clinicians were recruited from a broad range of memory clinics, located at one academic hospital, five non-academic teaching hospitals and one non-academic, non-teaching hospital. The mean duration of the post-diagnostic testing consultation was 19±8 minutes (M±SD).

Table 2 provides an overview of the diagnostic tests that patients underwent and whether individual test results were communicated to be abnormal, normal, or unclear by the clinician. Categorized communication behavior is displayed in Table 3 (concerning the MCI label, cause of symptoms, and prognostic information), and Table 4 (next steps in (care) planning). We allocated a unique ID number to each patient, and included those ID numbers in all tables in such a way that each row represents a patient.

Overall, qualitative analysis of the audiotaped post-diagnostic testing consultations revealed variation in clinicians' communication regarding the MCI label, the underlying cause of symptoms, prognostic information, and (care) planning. Based on further inspection of these results and relevant transcripts we formulated six core findings that comprise the essence of our results.

### Clinicians differed in the way they informed about the MCI label

Clinicians used the term MCI in approximately half of the consultations (Table 3 section A, and see also representative quotes 1.1 and 1.2 below). They also differed in whether they addressed the three MCI criteria to explain the patient's situation, i.e., clinicians often, yet not always, mentioned that the patient showed cognitive impairment and emphasized that the patient was not (yet) demented. In some consultations clinicians mentioned that there was no interference with daily living. The following two quotes illustrate how patients were provided

**Table 2. Clinicians' communication concerning results of diagnostic tests that patients underwent.**

| Patient ID | Neuropsychological assessment | Imaging (CT/MRI) | CSF by lumbar puncture | Amyloid PET | Visual aid used? |
|---|---|---|---|---|---|
| 1 | Abnormal | Unclear | Not performed | Results not yet available | CT/MRI images |
| 2 | Unclear | Normal | Abnormal | Not performed | No |
| 3 | Abnormal | Abnormal | Not performed | Not performed | No |
| 4 | Unclear | Normal | Not performed | Not performed | CT/MRI images |
| 5 | Abnormal | Abnormal | Not performed | Not performed | CT/MRI images |
| 6 | Not performed | Not performed | Not performed | Not performed | A drawing |
| 7 | Abnormal | Unclear | Not performed | Not performed | CT/MRI images |
| 8 | Unclear | Normal | Not performed | Not performed | CT/MRI images |
| 9 | Unclear | Abnormal | Normal | Not performed | CT/MRI images |
| 10 | Abnormal | Normal | Normal | Not performed | No |
| 11 | Normal | Abnormal | Abnormal | Not performed | CT/MRI images |
| 12 | Unclear | Normal | Not performed | Not performed | CT/MRI images |
| 13 | Abnormal | Normal | Not performed | Not performed | CT/MRI images |

*Notes.* This table indicates: i) whether a patient underwent a diagnostic test (if not, then indicated as 'not performed'), ii) whether test results were communicated to be abnormal, normal, or unclear by the clinician. The 'unclear' category was used when clinicians communicated that test results were unclear/ambiguous, or when clinicians' communication about the test results was unclear. In addition, the last column indicates whether the clinicians used visual aids to support the conversation with the patient and care partner. CSF = Cerebrospinal Fluid, CT = Computer Tomography, MRI = Magnetic Resonance Imaging, PET = Positron Emission Tomography.

with different amounts of information about the MCI label. In the first quote (1.1), the clinician thoroughly explains the patient's diagnosis by mentioning MCI and addressing all three MCI criteria.

> **Quote 1.1 (Patient ID 7):** "We do not think you have dementia. If it really were dementia, then you would expect to see more disorders, more tasks that you could not perform well during the tests. And you would also expect it to cause many more problems with daily activities. What it comes down to: Slight memory problems. [. . .] This is called mild cognitive impairment in English, or MCI."

The second quote (1.2) illustrates how clinicians sometimes explained the MCI label less extensively. Here, the clinician only mentions that the patient shows cognitive impairment, without using the term MCI.

> **Quote 1.2 (Patient ID 12):** "We have more or less said that, and this is not a real diagnosis but a description of the clinical picture, that the test results show a condition that is characterized by slight comprehension problems, apparently . . ., and some trouble remembering new information. Yes, there is a term used to describe your symptoms, but I won't mention it because you'll just forget and it's not really that important."

## Clinicians often addressed the cause of patients' symptoms in a tentative manner

In the majority of consultations, we observed communication regarding the underlying cause (s) of the patient's symptoms, as shown in section B of Table 3. This varied from a short statement by the clinician to a more elaborate conversation. If cause was addressed, the clinician often expressed himself/herself in a tentative manner, i.e., by using words such as possible,

**Table 3. Clinicians' communication regarding MCI label, cause, symptoms development, and risk of dementia.**

| Patient ID | A) Clinicians' communication regarding the MCI label | | | | B) Clinicians' communication about the cause of symptoms | C) Clinicians' communication about the expected course of symptoms | | | D) Clinicians' communication about the risk of developing dementia | |
|---|---|---|---|---|---|---|---|---|---|---|
| | MCI term used? | 1. Cognitive impairment | 2. No interference daily living | 3. Not demented | Underlying cause addressed? | Progression addressed? | Improvement addressed? | Stabilization addressed? | Risk addressed? | How? Formats used |
| 1 | Yes | Yes | No | No | Yes; possible AD | No | No | No | Yes | Verbal label, percentage, fifty-fifty, with time frame |
| 2 | Yes | Yes | No | Yes | Yes; probable AD | Yes | No | Yes | No | NA |
| 3 | No | No | No | No | Yes; possible psych distress | No | No | Yes | No | NA |
| 4 | Yes | Yes | Yes | Yes | Yes; possible AD | Yes | No | Yes | Yes | Verbal label only |
| 5 | No | Yes | Yes | Yes | Yes; aging | No | Yes | No | No | NA |
| 6 | Yes | Yes | No | Yes | Yes; possible AD | Yes | No | Yes | Yes | Verbal label, percentage, framing, with time frame |
| 7 | Yes | Yes | Yes | Yes | No | Yes | No | Yes | Yes | Verbal label only |
| 8 | No | No | No | Yes | Yes; psych distress | Yes | Yes | No | No | NA |
| 9 | No | Yes | No | Yes | Yes; potential vascular | Yes | No | Yes | No | NA |
| 10 | Yes | Yes | Yes | Yes | Yes; inexplicit | No | No | No | No | NA |
| 11 | No | No | No | No | Yes; possible AD | Yes | No | No | No | NA |
| 12 | No | Yes | No | No | No | Yes | Yes | Yes | No | NA |
| 13 | Yes | Yes | Yes | Yes | Yes; possible ADHD | No | No | No | Yes | Verbal label, fifty-fifty |

*Notes*. In the medical record, the patient with ID 10 was labeled as an individual with 'objective cognitive disorder(s) without meeting the criteria for dementia' and the patient with ID 11 as an individual with 'prodromal AD'. All other patients were labeled in the medical record as individuals with MCI. AD = Alzheimer's Disease, NA = not applicable, MCI = Mild Cognitive Impairment.

probable, potential, and may(be). This was independent of whether or not biomarker testing had been performed and independent of the biomarker result (normal/abnormal). This tendency to keep all options open, is illustrated by the next two quotes (2.1 and 2.2).

**Quote 2.1 (Patient ID 2 –abnormal biomarker status):** "But now that we see that the proteins in the cerebrospinal fluid are abnormal, we think this *might* suggest Alzheimer's disease after all."

**Quote 2.2 (Patient ID 10 –normal biomarker status):** "The lumbar puncture shows no signs of Alzheimer's disease. [. . .] Does this also mean that we can rule out an underlying disease? No, unfortunately we cannot really say that. The future will sort of reveal this."

In case of AD as the potential underlying cause of the patient's symptoms, only one clinician explained the difference between Alzheimer's disease and dementia as illustrated below (quote 2.3).

**Table 4. Overview of next steps in healthcare and disease management that were addressed by the clinician.**

| Patient ID | Further diagnostic testing | Medication / Medical treatment | Follow-up at memory clinic | Lifestyle adjustment/ advice | Study/trial participation | Driving ability testing | Follow-up by general practitioner | Referral to other healthcare professionals |
|---|---|---|---|---|---|---|---|---|
| 1 | No | Yes, no medical treatment available | Yes | No | Yes | No | No | No |
| 2 | No | Yes, no medical treatment available | Yes | Yes | Yes | No | Yes | Yes, psychologist |
| 3 | No | Yes, medication for mood disorder | Yes | Yes | No | No | Yes | Yes, psychiatrist |
| 4 | Yes, not pursued | Yes, no medical treatment available | Yes | Yes | No | No | Yes | No |
| 5 | No | Yes, adjusting medication for thyroid disorder | No | No | No | No | Yes | Yes, case manager |
| 6 | Yes, not pursued | No | Yes | No | No | No | Yes | No |
| 7 | No | Yes, no medical treatment available, vitamin B1 supplementation | Yes | Yes | No | No | No | Yes, neuropsychologist |
| 8 | No | No | Yes | No | No | No | No | Yes, psychologist |
| 9 | No | Yes, treatment for hypertension and hypercholesterolemia | Yes | Yes | No | Yes | No | No |
| 10 | No | No | Yes | No | No | No | No | No |
| 11 | No | Yes, no medical treatment available | Yes | Yes | Yes | No | No | No |
| 12 | No | No | Yes | Yes | No | No | No | No |
| 13 | Yes, pursued | Yes, cholinesterase inhibitors | Yes | No | No | No | No | Yes, occupational therapist |

**Quote 2.3 (Patient ID 6)**: "I can then prove, for example, that you have Alzheimer's disease, but that this has not yet developed into Alzheimer's dementia. There is a difference. You can have Alzheimer's disease for 20 years before you get Alzheimer's dementia."

## Clinicians (implicitly) steered against further biomarker testing by an unbalanced presentation and phrasing of arguments

If biomarker testing was not yet performed, clinicians sometimes introduced the possibility of additional/further biomarker testing (Table 4, and see representative quotes below). Clinicians mentioned as the main reason for biomarker testing to demonstrate the presence of AD pathology as the underlying cause of the patient's symptoms. However, they minimized the potential benefits/impact of biomarker testing by emphasizing that an abnormal biomarker result would not have (major) implications (as illustrated by quote 3.1 below). Of note, that a normal biomarker result would increase the likelihood of another (potentially treatable) cause, such as psychiatric problems, was never presented as an argument for biomarker testing. In addition, clinicians implicitly discouraged further biomarker testing by using diminutives and referring to biomarker tests as 'tricks', as illustrated by quote 3.2.

**Quote 3.1 (Patient ID 4):** "But with further tests we will be able to determine whether or not you have Alzheimer's disease. With a spinal tap, a lumbar puncture. However, the question is whether that is advisable, whether it is of any use to you, whether it would result in treatment. This is not the case. It would be purely for diagnostic purposes as there is no treatment."

> **Quote 3.2 (Patient ID 6):** "We can *play all kinds of tricks*, including a PET scan or examining the cerebrospinal fluid. Those can then help me confirm, for example, that you have Alzheimer's disease."

## Few clinicians informed about the patient's (personalized) risk of developing dementia

Clinicians communicated about the patient's risk of developing dementia in some consultations, as shown in Table 3 section D. If risk was discussed, this was most often discussed in a verbal format (see quote 4.1 below) without the use of percentages, natural frequencies, reference class or time frame.

> **Quote 4.1 (Patient ID 4):** "Of course, this *could be* an early warning sign of Alzheimer's disease with a dementia syndrome."

Still, the following quote (4.2) illustrates how some clinicians explicitly and elaborately explained a probability. Of note, the communicated risk was always in general terms and never personalized, e.g., taking advantage of knowledge on demographic or clinical characteristics, or (ab)normal biomarker result.

> **Quote 4.2 (Patient ID 6):** "Half of these people will remain stable, and the symptoms of about half of them will worsen and progress to dementia [*verbal format*]. [. . .] Let's say that the likelihood that you develop dementia within 5 years [*time frame*] is 50% [*percentage; negative framing*]. But the likelihood that you remain stable is also 50% [*percentage; positive framing*]."

## Clinicians often emphasized potential symptom stabilization or improvement

As displayed in Table 3 section C, clinicians often provided prognostic information by addressing the expected course of patient's symptoms. If addressed, the likelihood of symptom progression was often counterbalanced by addressing the unpredictability of symptom development, emphasizing that symptoms could also improve or stabilize. The next two quotes illustrate this tendency to foster hope.

> **Quote 5.1 (Patient ID 9):** "Yes, it is uncertain how it will progress, whether it will worsen or remain stable. This is something the tests can't tell us. [. . .] Anyway, let's hope your condition remains stable."

> **Quote 5.2 (Patient ID 2):** "Alzheimer's disease is, of course, a condition that is progressive, so it gets worse. But I can't predict how quickly this will happen. Your condition could remain stable for a long time and then deteriorate slowly, or it could progress more quickly. I can't predict that. [. . .] Yes, it may well remain stable for quite some time. Yes, that is quite possible."

## Clinicians informed about *current* next steps, but did not discuss *long-term* (care) planning

Table 4 shows that all clinicians discussed one or more next steps in care or symptom management, mostly medication. Regarding long-term (care) planning, clinicians only addressed

follow-up at the memory clinic to 'keep an eye on things', but in none of the consultations clinicians engaged in a conversation on *future* care or other *long-term* planning, e.g. concerning legal or financial issues.

Many clinicians provided general advise on how to manage/cope with the *current* cognitive symptoms, emphasizing what patients and their care partners could do themselves, e.g., concerning life style (quote 6.1 below) or the interaction between patient and care partner (quote 6.2).

**Quote 6.1 (Patient ID 7):** In general, it has been shown that a healthy lifestyle is really very important for the whole body, including the brain. In terms of food, i.e. alcohol and smoking, and in terms of exercise. Memory training exercises have also been looked into, but there is not a specific test or exercise that helps."

**Quote 6.2 (Patient ID 8):** "In general, I think it is important that the memory problems do not lead to too much irritation and frustration. But that is easier said than done, of course. So you must really do your best if you [patient] think that she is saying something important; try to really concentrate and remember it. And for you [partner], make sure that you choose the right moment to say something, just the two of you in a quiet room, with no TV or radio on, and clearly say: I really want you to remember what I'm telling you now. And repeat what you have to say a few times. These things actually always help."

In addition, some clinicians informed on how to treat or cope with *current* psychological/psychiatric problems, by means of medication or referral to a mental health professional, as illustrated by the next quote (6.3).

**Quote 6.3 (Patient ID 2):** "As far as counseling and guidance is concerned, I can't offer you much help from here. However, if you feel like, this message has come as a real shock, I could do with some support, then we do have a medical psychologist here who could help you and give you some extra guidance. If you are interested, just let us know. I can then arrange this for you. It is always difficult to gauge how you will react to the news. It is possible that once you are home you have a lot of difficulty coping with the situation."

## Discussion

In this observational study we aimed to explore memory clinic clinicians' communication during post-diagnostic testing consultations with MCI patients and their care partners, resulting in six core findings. First, we showed that approximately half of the clinicians used the term MCI, and they did not consistently use the three MCI criteria [5] to explain MCI. Second, clinicians often addressed the cause of patients' symptoms in a tentative manner, leaving the options open as it were, even when biomarker information was available and independent of biomarker status. Third, if biomarker testing was not yet performed, clinicians seemed to steer towards the decision *against* further testing. Fourth and fifth, few clinicians provided specific or personalized information on the risk to develop dementia, but many emphasized potential symptom stabilization or improvement. Finally, all clinicians advised on one or more short-term next steps in symptom management, but they rarely addressed long-term (care) planning topics.

Our coding of the observed communication was partially based on the recently updated American Academy of Neurology (AAN) practice guideline [3]. Despite the challenges regarding the use of the MCI label, this practice guideline explicitly recommends to discuss the MCI

diagnosis with patients and their care partners, to carefully educate them about the cause of symptoms and prognosis, and to encourage participation in long-term planning including living wills, finances and advance directives [3]. Discussing MCI with patients and their care partners might cause relieve, enabling them to label the patient's condition [16, 17]. Furthermore, an accurate MCI diagnosis might enable patients and care partners to be more engaged in healthcare and disease management and to prepare for the future [18]. However, the MCI label should then be communicated to them. Our results showed that clinicians do not consistently inform about MCI in practice. This is in line with previous research, where clinicians indicated to find it difficult to present MCI as a diagnosis, as its implications for the individual are quite uncertain [15]. Nevertheless, variation between clinicians with regard to disclosure strategies may not be unique to MCI [19], and is also reflected in their diagnostic communication with patients with dementia and their care partners [20].

Knowledge about the underlying cause of patients' symptoms could offer clinicians the opportunity to provide specific, more personalized, information about an individual patient's prognosis and, consequently, future (care) planning. Most patients and care partners also have a strong need to understand the patient's symptoms, and for information on the prognosis and implications [14]. Grill and colleagues state in their practice guideline that amyloid testing may offer such information [8]. Their point of view is more progressive then what is recommended in the AAN [3] or the Dutch MCI practice guidelines [21]. The Dutch guideline states that there are no accepted biomarkers available at this time to predict progression to dementia in patients with MCI, but stimulates clinicians to use diagnostic testing as a means to confirm alternative causes of the patient's cognitive decline that could be treatable, such as depression. Nonetheless, biomarker evidence could establish support for the underlying etiology (AD or non-AD, e.g., psychiatric problems) [5, 22]. A negative biomarker status may be especially informative, since it increases the likelihood of a, potentially treatable, etiology other than AD [23]. Still, in our sample, amyloid testing was infrequently used and did not have major consequences for clinicians' communication. That is, both negative and positive biomarkers results gave rise to a rather cautious discussion about the underlying cause of symptoms, and neither positive nor negative results were used to personalize the provision of information.

In fact, the patient's prognosis was only discussed in general terms, whereby the clinicians emphasized that symptoms may well remain stable or improve over time. Clinicians thus seemed to use this uncertainty to foster hope, by especially highlighting the possibility that positive outcomes could occur [24, 25]. This observation may also illustrate how clinicians try to balance the seemingly contrasting needs that could exist among patients, i.e., the needs for honest information versus optimism [26, 27]. However, since many patients and care partners prefer to reduce uncertainty about the diagnosis and prognosis to allow for preparation and future (care) planning, and report unmet information needs [12, 14, 28], clinicians' current approach may not match with patients' and care partners' needs. Information on prognosis could be personalized by using demographic (e.g. age) or clinical information (e.g. cognitive testing). In addition, ancillary investigations such as MRI, PET or CSF biomarkers could be used. Thus, even though biomarker testing does not *necessarily* decrease uncertainty, it still could provide patients and care partners with desired information about the future, even if the future is uncertain [29]. To attune to the individual's needs and preferences, clinicians, patients and care partners could discuss beforehand what potential test results might yield, and if biomarker testing is the preferred action at this point in time. Thus, the decision for or against testing, and which tests to use, should preferably be a shared decision [8, 30]. Our results however suggest that clinicians who addressed the possibility of further biomarker testing, did not present all arguments for biomarkers testing. Clinicians did communicate that a positive biomarker result would not impact any clinical decisions, as treatment is not available. From

literature however, it becomes increasingly clear that biomarkers have a particular good negative predictive value and may in turn increase or decrease the likelihood of another etiology, such as psychiatric problems (with potential treatment options) [3, 23, 31]. However, in our study, clinicians never mentioned this as an argument for additional biomarker testing.

Although the available practice guidelines recommend on *what* topics should be discussed when communicating about MCI, those guidelines do not provide clinicians with practical advise on *how* to optimally communicate these. To this background, clinicians have increasing amounts of (uncertain) information at their disposal that they could or should share, but they may not know how to meaningfully convey this information to their MCI patients. In addition, some clinicians might not be aware of the clinical value of the latest recommendations, since some of the recently published international frameworks are intended for research purposes [22]. Moreover, interpretation of biomarker results to gain estimates of the individual risk to develop dementia is not straightforward. Prediction models and computer-assisted clinical decision support systems may aid the clinician in interpreting clinical data and biomarker values and providing evidence-based individually-tailored prognostic information [23, 32, 33]. Nonetheless, even with a precise risk estimation in hand, the communication of this risk and its (uncertain) implications for the patient's future remains challenging [34], particularly so in cognitively impaired individuals. Therefore, communication-skills training teaching strategies and skills crucial for uncertainty communication and/or shared decision-making could also support clinicians to enhance their communication in clinical practice and, consequently, better attune to the individual needs of patients and their care partners.

Among the strengths of this study is its multicenter design, resulting in a heterogeneous group of clinicians and patients, contributing to the generalizability of the results. Moreover, we followed good practice in qualitative research [13], including double-coding of the data. However, some limitations deserve mentioning. First, the sample size of the current study was small. This study included all MCI patients (15%) from a larger observational study which included consecutive newly-referred memory clinic patients [12]. Compared to other (inter) national memory clinics [35, 36] and looking at the general population [37], this percentage of MCI patients appears very similar, increasing the likelihood that our sample is representative of the population. Still, sample bias might have occurred among participating clinicians (recruited based on self-selection), i.e., clinicians who are relatively comfortable with communication research and perhaps more skilled. In addition, our sample comprised more MCI patients seen by neurologists than geriatricians. Perhaps MCI patients are typically younger and therefore more often seen by a neurologist, instead of a geriatrician. In addition there might also be a clinician-induced bias, i.e., geriatricians may less often diagnose patients with MCI, for example because it is more difficult to determine objective cognitive decline in older patients. Since this was an explorative study, future research should investigate in a larger sample: 1) what factors could explain the variation in/are associated with clinicians' communication, e.g., consult duration, patient characteristics (for example the suspected underlying aetiology of the MCI), and/or clinician characteristics (such as clinicians' medical specialty), and; 2) the impact/consequences of different communication strategies on patients and their care partners, in terms of quality of life, emotional well-being, and (their engagement in) health and disease management. Based on that evidence, we could (further) develop recommendations on *how* to optimally disclose an MCI diagnosis and prognosis, and interventions to support clinicians in their adherence to MCI guidelines. In addition, investigating the recall and/or understanding of patients and care partners of the information provided during consultations would also be an interesting avenue for future investigation.

## Conclusion

The term MCI was used in half of the post-diagnostic testing consultations. Although variation existed, patients and care partners were often not provided with specific information about the underlying cause, risk to develop dementia and implications for long-term (care) planning. Since most patients and care partners have a strong need to understand the patient's symptoms, and for information on the prognosis and implications for the future, clinicians' current approach may not match with those needs.

## Supporting information

**S1 Appendix. The study-specific coding scheme.**
(DOCX)

## Acknowledgments

We thank Marleen Kunneman, Laxsini Murugesu, Bahar Azizi, Anneke Hellinga, and Sonja van Gils for their assistance with/during study design, data collection and/or coding.

## Author Contributions

**Conceptualization:** Leonie N. C. Visser, Femke H. Bouwman, Wiesje M. van der Flier, Ellen M. A. Smets.

**Data curation:** Leonie N. C. Visser, Ingrid S. van Maurik.

**Formal analysis:** Leonie N. C. Visser, Ingrid S. van Maurik, Ellen M. A. Smets.

**Funding acquisition:** Wiesje M. van der Flier, Ellen M. A. Smets.

**Investigation:** Leonie N. C. Visser, Ingrid S. van Maurik, Wiesje M. van der Flier, Ellen M. A. Smets.

**Methodology:** Leonie N. C. Visser, Ingrid S. van Maurik, Wiesje M. van der Flier, Ellen M. A. Smets.

**Project administration:** Leonie N. C. Visser, Ingrid S. van Maurik, Salka Staekenborg, Ralph Vreeswijk, Liesbeth Hempenius, Marlijn H. de Beer, Gerwin Roks, Leo Boelaarts, Mariska Kleijer, Wiesje M. van der Flier.

**Resources:** Wiesje M. van der Flier, Ellen M. A. Smets.

**Supervision:** Wiesje M. van der Flier, Ellen M. A. Smets.

**Validation:** Ellen M. A. Smets.

**Visualization:** Leonie N. C. Visser, Ingrid S. van Maurik.

**Writing – original draft:** Leonie N. C. Visser, Ingrid S. van Maurik.

**Writing – review & editing:** Leonie N. C. Visser, Ingrid S. van Maurik, Femke H. Bouwman, Salka Staekenborg, Ralph Vreeswijk, Liesbeth Hempenius, Marlijn H. de Beer, Gerwin Roks, Leo Boelaarts, Mariska Kleijer, Wiesje M. van der Flier, Ellen M. A. Smets.

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
