## [Decision Letter · Decision Letter 0]

27 Sep 2019

PONE-D-19-25712

Clinicians’ communication with patients receiving a MCI diagnosis: the ABIDE project.

PLOS ONE

Dear Dr. Visser,

Thank you for submitting your manuscript to PLOS ONE. After careful consideration by 2 Reviewers and an Academic Editor, all of the critiques of both Reviewers must be addressed in detail in a revision to determine publication status. If you are prepared to undertake the work required, I would be pleased to reconsider my decision, but revision of the original submission without directly addressing the critiques of the two Reviewers does not guarantee acceptance for publication in PLOS ONE. If the authors do not feel that the queries can be addressed, please consider submitting to another publication medium. A revised submission will be sent out for re-review. The authors are urged to have the manuscript given a hard copyedit for syntax and grammar.

**Comments to the Author**

1. Is the manuscript technically sound, and do the data support the conclusions?

Reviewer #1: Partly

Reviewer #2: Yes

2. Has the statistical analysis been performed appropriately and rigorously? 

Reviewer #1: N/A

Reviewer #2: Yes

3. Have the authors made all data underlying the findings in their manuscript fully available?

Reviewer #1: No

Reviewer #2: Yes

4. Is the manuscript presented in an intelligible fashion and written in standard English?

Reviewer #1: Yes

Reviewer #2: Yes

5. Review Comments to the Author

Reviewer #1: Overall this is a potentially very useful study for researchers and practitioners in the area of MCI research. I'm not aware of any studies to date that have examined clinical consultations for applications of recommended best practice, so this could be a valuable addition to the literature. It was appropriate in this case not to make the data fully available as per PLOS policy, given the nature of the data (clinical consultations). However I also think there are some important ways the manuscript could be improved.

1. Diagnostic process and categories: Could the authors explain a little more about the diagnostic pathway for MCI in the Netherlands? It looks as though all patients were recruited at the point when the MCI diagnosis was first to be delivered to them, is this correct? Would all the patients have had the same battery of tests to confirm AD/ dementia/ MCI status? Additionally, two of the patients were assigned different clinical labels - prodromal AD and objective cognitive disorders without dementia. I would expect this to affect the sorts of discussions the clinicians had with patients (especially since prodromal AD suggests a more known prognosis than MCI, which may not develop into AD)

2. Recruitment/ sampling: The sample of consultations is quite small at N=13. Quality over quantity in qualitative research, fair enough. But how was the sample size decided on, and what recruitment approach was used?

3. Data presentation: Generally I found the Tables in the manuscript fairly helpful. The one exception is Table 5. Personally, I'd prefer these 'raw' qualitative data included in the flow of the text to illustrate the findings as one reads along. My other point here is that seeing some of the contrasts in the data would help. Eg. Quote 1.1. - the clinician explicitly uses MCI, but in half the cases they did not - what happened in such cases?

4. Data analysis - an a priori coding framework was used, based on current guidelines. This seems like a sensible approach. Was there provision for 'bottom-up' coding of any relevant data that didn't fit the categories? Why/ why not?

5. Clinical implications: In the discussion, there is some implicit criticism of current clinical practice which I think may be a little unfair. I have in mind especially the suggestion to 'clearly communicate' the prognosis (p. 15) - given the uncertainty of prognosis, as you then go on to acknowledge, I think this is a real challenge for clinicians to provide good information. Should risk be positively or negatively framed? Given as absolute or relative risks? etc? This is a really complex issue and clearly, further work is needed to understand what risk information needs giving to which patients, in what format.

I also think there is some nuance to add to the points about fostering hope on p. 16. Here I think clinicians are negotiating some tricky, contrasting needs - on the one hand, patients want clinicians to communicate honestly, but often also desire optimism from clinicians (this is quite an old paper now, but illustrates the point nicely: Kutner, J. S., Steiner, J. F., Corbett, K. K., Jahnigen, D. W., & Barton, P. L. (1999). Information needs in terminal illness. Social science & medicine, 48(10), 1341-1352.). Maybe the point here is to improve communication about lifestyle modifications that may have a chance of improving prognosis.

6. Finally, the focus on clinicians' communication was helpful, but I wonder what the patients made of the communication? Were they passive in accepting the clinicians' words? Did they ask further questions if/ when needed? Did they seem to understand their prognosis? Where information was provided on causes, was this proactively given by the clinician or sought by the patient? Etc. I think there is some really useful info you'll have here that could considerably add to the interest and importance of the article. To take just one example, quote 1.2 in Table 5 is a genuinely bad example of clinical communication - what did the patient do in response?

Reviewer #2: This is a nice paper on an important topic that has received surprisingly little attention—diagnostic disclosure of a Mild Cognitive Impairment (MCI) diagnosis. The authors present data from a subset of participants in a large qualitative study, providing an in depth assessment of the delivery of MCI diagnosis in 13 patient interactions (by 7 different clinicians). They observe heterogeneity in diagnostic delivery, including some surprising things that are frequently missing from clinical interactions like discussion of long-term prognosis (risk for dementia) and long-term planning. Yet, the sample is small. And it is quite possible that important trends (e.g., 4 out of 5 exchanges where the presumed etiology was Alzheimer’s disease did include discussion of potential worsening over time [presumably to dementia]) were not fully fleshed out. There is also a question of selection bias that must be addressed. Altogether, this paper is an important contribution and should sound an alarm for clinicians and professional organizations alike, to use formal or informal checklists of the essential topics that should be addressed when delivering the MCI diagnosis.

Abstract

• Should include the physician specialties

• The first conclusion in the abstract is that information provided to patients and families about prognosis could be more personalized. But could it? While some risk estimates exist, are these “ready for primetime?” At best, they face tremendously limitations in generalizability.

• Alternatively, perhaps the point above about 4 out of 5 exchanges with patients for whom the presumed etiology was Alzheimer’s disease including discussion of risk for progression indicates that more personalized assessments are being performed for a subset of patients.

Introduction

• Well written and pithy. It may make sense (here or elsewhere) to give additional context for the study. This might briefly include important information about national practice guidelines (e.g., approved therapies, reimbursement for biomarkers, etc.)?

Methods

• There is no discussion of saturation or how the number of participants or interviews were determined

• Similarly, there is imbalance in the number of interviews from neurologists vs. geriatricians

• Presumably these are a subset of patients from a larger study of diagnostic exchange (Visser et al. Alzheimer's & Dementia: Diagnosis, Assessment & Disease Monitoring. 2019). Yet, the selection does not appear to be as simple as all MCI diagnoses from the previous study (n=21). How then were these cases selected and what risk of bias comes with the selection? The paper would be strengthened if all MCI cases from the previous study were used.

Results

• The heading for section 3.2 is unclear

Discussion

• The duration of the consult (Table 1) seems worth comment.

6. PLOS authors have the option to publish the peer review history of their article (what does this mean?). If published, this will include your full peer review and any attached files.

**Do you want your identity to be public for this peer review?** For information about this choice, including consent withdrawal, please see our Privacy Policy.

Reviewer #1: No

Reviewer #2: Yes: Joshua D. Grill

We would appreciate receiving your revised manuscript by February, 2020. To enhance the reproducibility of your results, we recommend that if applicable you deposit your laboratory protocols in protocols.io, where a protocol can be assigned its own identifier (DOI) such that it can be cited independently in the future. For instructions see: http://journals.plos.org/plosone/s/submission-guidelines#loc-laboratory-protocols

We look forward to receiving your revised manuscript.

Kind regards,

Stephen D. Ginsberg, Ph.D.

Section Editor

PLOS ONE
---

## [Author Response · Author response to Decision Letter 0]

7 Dec 2019

Review Comments to the Author

Reviewer #1: Overall this is a potentially very useful study for researchers and practitioners in the area of MCI research. I'm not aware of any studies to date that have examined clinical consultations for applications of recommended best practice, so this could be a valuable addition to the literature. It was appropriate in this case not to make the data fully available as per PLOS policy, given the nature of the data (clinical consultations). However I also think there are some important ways the manuscript could be improved.

1. Diagnostic process and categories: 

- Could the authors explain a little more about the diagnostic pathway for MCI in the Netherlands? It looks as though all patients were recruited at the point when the MCI diagnosis was first to be delivered to them, is this correct? 

We selected MCI patients from the larger ABIDE observational study, where audio-recordings were made of the routine clinician-patient consultations with 136 newly referred patients during their diagnostic work up (for a detailed description, see Visser et al. (1)). In the Netherlands, patients are referred to the memory clinic by another medical doctor, mostly their general practitioner, i.e. no self-referral. Of the total sample of patients participating in the larger ABIDE study, 21 (15%) were diagnosed with mild cognitive impairment (MCI), based on retrospectively obtained medical record data. Fully audio-recorded post-diagnostic testing consultations were available for 13/21 MCI patients. These audio-recorded consultations comprise the data used for the current manuscript.

We agree with the reviewer that we could have explained this more clearly, and now elaborated a bit further in the method section 2.2 of our manuscript on page 6: 

“All clinicians involved in diagnostic consultations at seven Dutch memory clinics were eligible and invited to participate (self-selection). The newly referred patients of consenting clinicians (in the Netherlands patients are referred to the memory clinic by another medical doctor, i.e. no self-referral) and accompanying care partners were invited to participate. Next, audio-recordings were made of the clinician-patient consultations when patients were seen as part of normal clinical care at the memory clinic. For the current study we selected fully audio-recorded post-diagnostic testing consultations with MCI patients (N=13), as based on their retrospectively obtained medical records.”

In addition, we added Figure 1, a flow diagram, on page 7 with the following note:

“Note. Of the total sample of patients participating in the ABIDE observational study (1), 21 (15%) were diagnosed with mild cognitive impairment (MCI), based on their retrospectively obtained medical record data. Post-diagnostic testing audio recordings were available from 13/21 MCI patients, who were seen by ten clinicians in seven Dutch memory clinics.”

- Would all the patients have had the same battery of tests to confirm AD/dementia/MCI status? 

No efforts were made to standardize the diagnostic work up between memory clinics or clinicians, since our aim was to observe and describe naturalistic clinician-patient communication during routine diagnostic clinical consultations in a wide range of memory clinics. This resulted in variation between patients in diagnostic tests used; some memory clinics had a fixed diagnostic care path, i.e., they used a standard diagnostic protocol in which the decisions regarding diagnostic testing were already made on a memory clinic level (what tests to use and in which order), in other memory clinics, decisions about testing were made at a patient level (2). We added the diagnostic tests that each patient underwent, and the communicated test results in Table 2. In addition, we now elaborated on this in the method section on page 6:

“Since our aim was to observe and describe naturalistic clinician-patient communication during the routine diagnostic work up for dementia in a broad range of memory clinics, no efforts were made to intervene or standardize the diagnostic work up (for a detailed description see Visser et al., (1) and Table 2 for details on diagnostic tests that patients underwent).”

- Additionally, two of the patients were assigned different clinical labels - prodromal AD and objective cognitive disorders without dementia. I would expect this to affect the sorts of discussions the clinicians had with patients (especially since prodromal AD suggests a more known prognosis than MCI, which may not develop into AD)

The reviewer addresses an interesting point. Patient ID 11 was labeled with prodromal AD. The coded findings for this patients (Table 3) show that the clinician addressed progression of symptoms, and that this was the only patients where this statement was not counterbalanced with a statement on possible stabilization of symptoms. For this patient, one might expect the clinician to communicate in more details about this patients’ risk of developing a dementia syndrome in the future. However, such a discussion on risk of developing dementia was not observed in this patient. Patient ID 10 was labeled with objective cognitive disorder without dementia in the medical record. But in the clinical consultation, the diagnosis for this patient was clearly communicated as an MCI label (Table 3). Our sample is however too small to make sensible conclusions based on these observations. For the reader to appreciate possible differences in communication, we now included a note underneath Table 3, stating which patient had the prodromal AD label (patient ID 11) and which patient was labeled with objective cognitive disorder without dementia (patient ID 10).

2. Recruitment/ sampling: The sample of consultations is quite small at N=13. Quality over quantity in qualitative research, fair enough. But how was the sample size decided on, and what recruitment approach was used?

The MCI patients were selected from a larger study that investigated naturalistic clinician-patient communication during the diagnostic work-up for dementia, in the context of routine clinical care. In this larger study, all newly referred patients and accompanying care partners were invited to participate. Of the total sample of patients participating in the larger observational study, 21 (15%) were diagnosed with mild cognitive impairment (MCI), as based on retrospectively collected medical record data. Fully audio-recorded post-diagnostic consultations were available for 13/21 MCI patients. We have now added this to the method section (2.2 Sample, procedure and questionnaires, page 6.) and included a flow diagram to describe this patient flow (Figure 1, page 7). 

3. Data presentation: 

- Generally I found the Tables in the manuscript fairly helpful. The one exception is Table 5. Personally, I'd prefer these 'raw' qualitative data included in the flow of the text to illustrate the findings as one reads along. 

We have now included the quotes in the main text as suggested, and excluded Table 5. 

- My other point here is that seeing some of the contrasts in the data would help. Eg. Quote 1.1. - the clinician explicitly uses MCI, but in half the cases they did not - what happened in such cases?

Throughout the manuscript we have aimed to show contrasts by means of representative quotes. An example of what happened in the cases where a clinician did not explicitly use MCI, is shown in Quote 1.2. Here, the clinician only mentions that the patient shows cognitive impairment, without using the term MCI.

Quote 1.2 (Patient ID 12): “So the point is that I find it hard to explain how we can clarify this. The tests are also not conclusive, so yes... [...]. We have more or less said that, and this is not a real diagnosis but a description of the clinical picture, that the test results show a condition that is characterized by slight comprehension problems, apparently.... and some trouble remembering new information. Yes, there is a term used to describe your symptoms, but I won't mention it because you'll just forget and it's not really that important.”

Another example of contrasts in the data are for example shown in section 3.4. The first quote illustrates that if risk was discussed, this was most often discussed in a verbal format without the use of percentages, natural frequencies, reference class or time frame. 

Quote 4.1 (Patient ID 4): “Of course, this could be an early warning sign of Alzheimer's disease with a dementia syndrome.”

On the other hand, some clinicians explicitly and elaborately explained a probability, shown by the following quote: 

Quote 4.2 (Patient ID 6): “Half of these people will remain stable, and the symptoms of about half of them will worsen and progress to dementia [verbal format]. [...] Let's say that the likelihood that you develop dementia within 5 years [time frame] is 50% [percentage; negative framing]. But the likelihood that you remain stable is also 50% [percentage; positive framing].”

That we aimed to show contrasts in the data, might have been unclear as all quotes were previously summarized in table 5. We have now included the quotes in the main text, and we now better refer to those representative quotes, which hopefully sufficiently illustrates the contradictions in the qualitative data. 

4. Data analysis - an a priori coding framework was used, based on current guidelines. This seems like a sensible approach. Was there provision for 'bottom-up' coding of any relevant data that didn't fit the categories? Why/ why not?

First, we indeed used deductive coding based on codebook inspired by the current guidelines and previous research. In that coding scheme (see S1 Appendix), there was some room for bottom-up coding, e.g., by means of the category ‘other’ with regard to next steps in care / planning.

Second, we took an inductive/bottom-up approach, by using thematic content coding looking at all coded categories together with relevant transcripts/quotes, identifying six core findings/themes that emerged from the data. We now better clarified this in the method section (2.3) on page 8:

“The final codes combined with the relevant transcribed segments of the audio-recordings were then inductively categorized further by LNCV, ISvM and EMAS, identifying core findings emerging from the data.”

5. Clinical implications: 

- In the discussion, there is some implicit criticism of current clinical practice which I think may be a little unfair. I have in mind especially the suggestion to 'clearly communicate' the prognosis (p. 15) - given the uncertainty of prognosis, as you then go on to acknowledge, I think this is a real challenge for clinicians to provide good information. Should risk be positively or negatively framed? Given as absolute or relative risks? etc? This is a really complex issue and clearly, further work is needed to understand what risk information needs giving to which patients, in what format.

We agree with the reviewer that prognostic information is often uncertain and that risks are difficult/challenging to communicate clearly, because of various reasons. That is, in fact the point we try to make in the paragraph on page 20. We thank the reviewer for pointing this out, since it is absolutely not our intention to criticize clinicians on this matter. We now changed our suggestion on page 15 and excluded prognosis from that statement, which now reads:

“However, the MCI label should then be communicated to them.”

- I also think there is some nuance to add to the points about fostering hope on p. 16. Here I think clinicians are negotiating some tricky, contrasting needs - on the one hand, patients want clinicians to communicate honestly, but often also desire optimism from clinicians (this is quite an old paper now, but illustrates the point nicely: Kutner, J. S., Steiner, J. F., Corbett, K. K., Jahnigen, D. W., & Barton, P. L. (1999). Information needs in terminal illness. Social science & medicine, 48(10), 1341-1352.). Maybe the point here is to improve communication about lifestyle modifications that may have a chance of improving prognosis.

This point is well taken, and we would like to thank the reviewer for this concrete suggestion about an alternative explanation of our observations. We now added the following sentence to our discussion on page 19:

“This observation may also illustrate how clinicians try to balance the seemingly contrasting needs that could exist among patients, i.e., the needs for honest information versus optimism (3, 4). However, since many patients and care partners prefer to reduce uncertainty about the diagnosis and prognosis to allow for preparation and future (care) planning (1, 5), clinicians’ current approach may not match with patients’ and care partners’ needs.”

We also tried to add some nuance to this discussion by emphasizing that many patients and care partners prefer information, and that clinicians’ approach may not match with their needs.

6. Finally, the focus on clinicians' communication was helpful, but I wonder what the patients made of the communication? Were they passive in accepting the clinicians' words? Did they ask further questions if/ when needed? Did they seem to understand their prognosis? Where information was provided on causes, was this proactively given by the clinician or sought by the patient? Etc. I think there is some really useful info you'll have here that could considerably add to the interest and importance of the article. To take just one example, quote 1.2 in Table 5 is a genuinely bad example of clinical communication - what did the patient do in response?

Findings from the larger ABIDE observational study (1) showed that memory clinic patients and care partners only rarely (2% of the consultations) expressed what kind or how much information they preferred, e.g., by asking questions or by a remark. The amount of uncertainty expressions by the clinician in these consultations was the scope of a recently published ABIDE paper, and showed that patients and care partners elicited, respectively, few (15% and 23%) of these uncertainty expressions, through questions (6). Together these findings show that patient and caregiver active involvement in diagnostic consultations at the memory clinic is limited. We do also know, as previously reported based on recent focus group data (5), survey data(1) and a Delphi consensus study(7), that patients and care partners report an unmet need for information, especially about the future and what the test results mean for their daily life.

As the reviewer states, in this current study we focused on clinicians’ communication with MCI patients and did not code MCI patients’ communication in response to the provided diagnosis or prognosis. After the post-diagnostic testing consultation we did aim to survey patients’ and care partners’ understanding of 1) the communicated diagnosis: did you/your relative receive a diagnosis (No or Yes, namely…), and 2) the communicated prognosis: did the clinician disclose the risk of developing Alzheimer’s dementia (No or Yes, namely…)? The majority of patients (8/13) and caregivers (10/13) reported that they had received a diagnosis, however only two patients and two caregivers stated MCI as the received diagnosis. However, the written answers on these questions by patients and care partners were often unclear or their answers to the open-ended questions (namely…?) were lacking completely. We do not know whether this was the result of patients and care partners not understanding/remembering the diagnosis/prognosis, or not understanding the questions or our question format. We were therefore not convinced of the validity/reliability of these data and decided not to incorporate these in the current manuscript. Nevertheless, studying the information recall and/or understanding of patients and care partners of the information provided during consultations would most certainly be an interesting avenue for future investigation. 

We now added this future research direction to the discussion on page 21:

“In addition, investigating the recall and/or understanding of patients and care partners of the information provided during consultations would also be an interesting avenue for future investigation.”

Reviewer #2: This is a nice paper on an important topic that has received surprisingly little attention—diagnostic disclosure of a Mild Cognitive Impairment (MCI) diagnosis. The authors present data from a subset of participants in a large qualitative study, providing an in depth assessment of the delivery of MCI diagnosis in 13 patient interactions (by 7 different clinicians). They observe heterogeneity in diagnostic delivery, including some surprising things that are frequently missing from clinical interactions like discussion of long-term prognosis (risk for dementia) and long-term planning. Yet, the sample is small. And it is quite possible that important trends (e.g., 4 out of 5 exchanges where the presumed etiology was Alzheimer’s disease did include discussion of potential worsening over time [presumably to dementia]) were not fully fleshed out. There is also a question of selection bias that must be addressed. 

With regard to these issues raised, we provided a point-by-point reply below.

Altogether, this paper is an important contribution and should sound an alarm for clinicians and professional organizations alike, to use formal or informal checklists of the essential topics that should be addressed when delivering the MCI diagnosis.

Abstract

• Should include the physician specialties

We’ve now included the physician specialties (eight neurologists and two geriatricians) in the abstract (page 3). 

• The first conclusion in the abstract is that information provided to patients and families about prognosis could be more personalized. But could it? While some risk estimates exist, are these “ready for primetime?” At best, they face tremendously limitations in generalizability.

In the abstract we provide a conclusion based on our six key findings, and stated that “Clinicians’ information provision about the underlying cause, prognosis and implications for long-term (care) planning in MCI could be more specific and personalized”. We did not intend to put emphasis on personalizing prognostic information (only), since we believe it is important to personalize all information provided, for example by adjusting the amount and content of information to the individual patient’s needs and preferences. From our previous publication(1) we know that patients’ and caregivers’ information preferences are however often not discussed / expressed in consultations. We do agree with the reviewer that the data/results included in the current manuscript better fit the conclusion that information provision could be more specific, and provide a less solid base for concluding that this information could be more personalized. The reviewer’s point is thus well-taken. We now deleted ‘personalized’ from the conclusion in the abstract and in the discussion.

• Alternatively, perhaps the point above about 4 out of 5 exchanges with patients for whom the presumed etiology was Alzheimer’s disease including discussion of risk for progression indicates that more personalized assessments are being performed for a subset of patients.

Although recent risk models are promising (8), we agree with the reviewer’s statements above that clinicians might be hesitant to provide precise risk estimates (percentages), and that the existing risk estimation models may not be ready (yet) for primetime in clinical practice. Nevertheless, we do believe that clinicians could use other formats than percentages to inform about the chance of developing dementia (e.g., a verbal label, such as ‘people with MCI have a higher chance of developing dementia’), and that this information is relevant for patients for whom the presumed etiology is Alzheimer’s disease, but also important for patients with other presumed underlying causes. Most patients and caregivers report that such information is very important to discuss during the diagnostic process (7), and many would have liked to receive more information about the future (1). We however observed that in most consultations with our MCI patients (8/13), the chance, likelihood, risk, possibility of developing dementia was not addressed, in fact, dementia was not mentioned at all. 

In addition, demographic, clinical and biomarker information could provide guidance to personalize prognostic information for both patients in which AD is the presumed etiology and others. E.g., in two out of the eight consultations (patient ID 2 and ID 11) in which the risk/possibility of developing dementia was not addressed, AD was communicated as a potential/probable cause of MCI, and biomarker results were positive. This information did not gave rise to a conversation about dementia, let alone to personalize information about the chance of dementia. In comparison to other patients with negative biomarker results, these patients have a high(er) chance to develop dementia. And if, for example, neuropsychological testing, anamnestic information, young age or negative biomarker results point in the direction of a psychiatric cause of MCI, developing dementia is less likely. Thus, even though demographic, clinical, or biomarker information could not (yet) result in a precise risk estimate, it still could provide patients and care partners with desired information about the future, even if the future is still uncertain.

We therefore formulated as key finding 4 (section 3.4) that in our sample, information provision about the patient’s risk of developing dementia could be more specific and personalized. Yet, we now added a future research direction in the discussion on page 21 reflecting to reviewer’s hypothesis that clinicians might personalize their assessments and consequently their communication, based on the suspected underlying etiology:

“Since this was an explorative study, future research should investigate in a larger sample: 1) what factors could explain the variation in/are associated with clinicians’ communication, e.g., consult duration, patient characteristics (for example the suspected underlying aetiology of the MCI), and/or clinician characteristics (such as clinicians’ medical specialty)”

Introduction

• Well written and pithy. It may make sense (here or elsewhere) to give additional context for the study. This might briefly include important information about national practice guidelines (e.g., approved therapies, reimbursement for biomarkers, etc.)?

We thank the reviewer for this concrete suggestions. In the discussion on page 19, we mentioned the Dutch (9) MCI-practice guidelines, and we now elaborated on this a bit further:

“The Dutch guideline states that there are no accepted biomarkers available at this time to predict progression to dementia in patients with MCI, but stimulates clinicians to use diagnostic testing as a means to confirm alternative causes of the patient’s cognitive decline that could be treatable, such as depression.”

We now also elaborated a bit more in the methods section on page 6 providing information about the Dutch referral/memory clinic system:

“The newly referred patients of consenting clinicians (in the Netherlands patients are referred to the memory clinic by another medical doctor, i.e. no self-referral) and accompanying care partners were invited to participate. Next, audio-recordings were made of the clinician-patient consultations when patients were seen as part of normal clinical care at the memory clinic. Since our aim was to observe and describe naturalistic clinician-patient communication during the routine diagnostic work up for dementia in a broad range of memory clinics, no efforts were made to intervene or standardize the diagnostic work up (for a detailed description see Visser et al., (1), and Table 2 for details on diagnostic tests that patients underwent).”

Methods

• There is no discussion of saturation or how the number of participants or interviews were determined

We could have been more specific about our sample selection and recruitment. We thank the reviewer for pointing this out. The sample size of this study was not based on saturation. The MCI patients were selected from a larger study that investigated clinician-patient communication during the diagnostic work-up for dementia. In this larger study, all newly referred patients and accompanying care partners were invited to participate (for a detailed description see (1)). Of the total sample of patients participating in the larger ABIDE study, 21 (15%) were diagnosed with mild cognitive impairment (MCI). Fully audio-recorded post-diagnostic consultations were available for 13/21 MCI patients. We have now added this to the method section (2.2 Sample, procedure and questionnaires, page 6.) and included a flow diagram to describe this patient flow (Figure 1, page 7). 

• Similarly, there is imbalance in the number of interviews from neurologists vs. geriatricians

The MCI patients in this sample are more often seen by neurologists (n=10) than by geriatricians (n=3). Since we did not aim to make a comparison based on specialism, we did not specifically set out to generate a sample of patients seen by a balanced number of neurologists versus geriatricians. But for the larger observational study we did set out to involve a heterogeneous sample of Dutch memory clinics, located at one academic hospital (20 patients, 12 clinicians), six non-academic teaching hospitals (106 patients; 27 clinicians) and one non-academic, non-teaching hospital (10 patients; 2 clinicians). The clinicians were recruited by self-selection, and 41 were willing to participate (42% neurologists, 32% geriatricians, 27% other profession (e.g., psychiatrists)). So, part of the imbalance might be caused by self-selection, whereby neurologists were more often willing to participate. 

However, it could also be that there are relatively more neurologists involved at memory clinics in the Netherlands than geriatricians. We don’t have any official numbers to support this hypothesis. Or, alternatively, regarding our selection of MCI patients, it could be that MCI patients are typically younger and are therefore more often referred to a neurologist, instead of a geriatrician. In addition, there might also be a physician induced bias as geriatricians may less often diagnose their older patients with MCI, for example because it is more difficult to determine objective cognitive decline. But also a delay may have occurred, as elderly patients (seen by geriatricians) may more often think that their mild (memory) complaints are a result of normal aging and may therefore less often seek medical help early on (in the stage of MCI). Since this was an explorative study, future research should investigate in a larger sample what factors could explain the variation in clinicians’ communication, e.g., patient or clinician characteristics such as clinicians’ medical specialty. We now elaborated further on this issue, in the discussion section on page 21 (see also our reply to the reviewer’s next point below, the highlighted added text).

• Presumably these are a subset of patients from a larger study of diagnostic exchange (Visser et al. Alzheimer's & Dementia: Diagnosis, Assessment & Disease Monitoring. 2019). Yet, the selection does not appear to be as simple as all MCI diagnoses from the previous study (n=21). How then were these cases selected and what risk of bias comes with the selection? The paper would be strengthened if all MCI cases from the previous study were used.

The reviewer is correct that this is a subset of the larger ABIDE observational study. We have now included a flow diagram (see Figure 1, page 7) to clarify the subject selection. From the 21 MCI patients, no fully recorded post-diagnostic consultations were available in n=8. This was for random reasons, e.g., because the clinician forgot to start the audio-recording or the post-diagnostic testing consultation was scheduled beyond our time window for data collection. In the discussion section on page 21 we address our small sample size and potential sampling bias:

“However, some limitations deserve mentioning. First, the sample size of the current study was small. This study included all MCI patients (15%) from a larger observational study which included consecutive newly-referred memory clinic patients (1). Compared to other (inter)national memory clinics (10, 11) and looking at the general population (12), this percentage of MCI patients appears very similar, increasing the likelihood that our sample is representative of the population. Still, sample bias might have occurred among participating clinicians (recruited based on self-selection), i.e., clinicians who are relatively comfortable with communication research and perhaps more skilled. In addition, our sample comprised more MCI patients seen by neurologists than geriatricians. Perhaps MCI patients are typically younger and therefore more often seen by a neurologist, instead of a geriatrician. In addition there might also be a clinician-induced bias, i.e., geriatricians may less often diagnose patients with MCI, for example because it is more difficult to determine objective cognitive decline in older patients. Since this was an explorative study, future research should investigate in a larger sample: 1) what factors could explain the variation in/are associated with clinicians’ communication, e.g., consult duration, patient characteristics (for example the suspected underlying aetiology of the MCI), and/or clinician characteristics (such as clinicians’ medical specialty”

Results

• The heading for section 3.2 is unclear

We have now changed this subheading to “3.2 clinicians often addressed the cause of patients’ symptoms in a tentative manner”, and hope that this adjustment clarifies its meaning (page 13). 

Discussion

• The duration of the consult (Table 1) seems worth comment.

We observed some variation in consult duration. However, our sample is too small to analyze differences in communication based on consult duration, or to analyze correlations between consult duration and patient or clinician characteristics. The duration of the post-diagnostic testing consultation in this sample of MCI patients (M= 19, SD= 8 minutes), does not appear to deviate from the larger ABIDE sample, including patients with dementia and without objective cognitive decline (M= 19, SD= 13 minutes). However, we are not sure if the reviewer means that the variation in consult duration is worth comment, or that the duration in itself seems worth a comment (does the reviewer consider the consultation to be long or short?). We now added looking into consult duration as a future research direction, in the discussion section on page 21 (see below). If needed, we are happy to elaborate further on this issue.

“Since this was an explorative study, future research should investigate in a larger sample: 1) what factors could explain the variation in/are associated with clinicians’ communication, e.g., consult duration, …”

 

References in this revision note.

1. Visser LNC, Kunneman M, Murugesu L, van Maurik I, Zwan M, Bouwman FH, et al. Clinician-patient communication during the diagnostic workup: The ABIDE project. Alzheimer's & Dementia: Diagnosis, Assessment & Disease Monitoring. 2019;11:520-8.

2. Kunneman M, Bouwman FH, Smets EMA, van der Flier WM. Diagnostiek van dementie: praktijkvariatie in Nederlandse geheugenpoliklinieken. Neuropraxis. 2018;22(5):137–46.

3. Lakin JR, Jacobsen J. Softening Our Approach to Discussing Prognosis. JAMA Intern Med. 2019;179(1):5-6.

4. Kutner JS, Steiner JF, Corbett KK, Jahnigen DW, Barton PL. Information needs in terminal illness. Soc Sci Med. 1999;48(10):1341-52.

5. Kunneman M, Pel-Littel R, Bouwman FH, Gillissen F, Schoonenboom NSM, Claus JJ, et al. Patients' and caregivers' views on conversations and shared decision making in diagnostic testing for Alzheimer's disease: The ABIDE project. Alzheimers Dement (N Y). 2017;3(3):314-22.

6. Visser LNC, Pelt SAR, Kunneman M, Bouwman FH, Claus JJ, Kalisvaart KJ, et al. Communicating uncertainties when disclosing diagnostic test results for (Alzheimer's) dementia in the memory clinic: The ABIDE project. Health Expect. 2019.

7. Fruijtier AD, Visser LNC, van Maurik IS, Zwan MD, Bouwman FH, van der Flier WM, et al. ABIDE Delphi study: topics to discuss in diagnostic consultations in memory clinics. Alzheimers Res Ther. 2019;11(1):77.

8. van Maurik IS, Vos SJ, Bos I, Bouwman FH, Teunissen CE, Scheltens P, et al. Biomarker-based prognosis for people with mild cognitive impairment (ABIDE): a modelling study. Lancet Neurol. 2019.

9. Richtlijn diagnostiek en behandeling van dementie - Addendum Mild Cognitive Impairment (MCI). Utrecht: Kennisinstituut van de Federatie Medisch Specialisten; 2017.

10. van der Flier WM, Scheltens P. Amsterdam Dementia Cohort: Performing Research to Optimize Care. J Alzheimers Dis. 2018;62(3):1091-111.

11. Roberts R, Knopman DS. Classification and epidemiology of MCI. Clin Geriatr Med. 2013;29(4):753-72.

12. Association As. 2019 Alzheimer’s disease facts and figures. 2019.

---

## [Decision Letter · Decision Letter 1]

17 Dec 2019

Clinicians’ communication with patients receiving a MCI diagnosis: the ABIDE project.

PONE-D-19-25712R1

Dear Dr. Visser,

We are pleased to inform you that your manuscript has been judged scientifically suitable for publication and will be formally accepted for publication once it complies with all outstanding technical requirements.

With kind regards,

Stephen D. Ginsberg, Ph.D.

Section Editor

PLOS ONE

**Comments to the Author**

1. If the authors have adequately addressed your comments raised in a previous round of review and you feel that this manuscript is now acceptable for publication, you may indicate that here to bypass the “Comments to the Author” section, enter your conflict of interest statement in the “Confidential to Editor” section, and submit your "Accept" recommendation.

Reviewer #1: All comments have been addressed

Reviewer #2: All comments have been addressed

2. Is the manuscript technically sound, and do the data support the conclusions?

Reviewer #1: Yes

Reviewer #2: Yes

3. Has the statistical analysis been performed appropriately and rigorously? 

Reviewer #1: N/A

Reviewer #2: Yes

4. Have the authors made all data underlying the findings in their manuscript fully available?

Reviewer #1: Yes

Reviewer #2: Yes

5. Is the manuscript presented in an intelligible fashion and written in standard English?

Reviewer #1: Yes

Reviewer #2: Yes

6. Review Comments to the Author

Reviewer #1: (No Response)

Reviewer #2: The authors have thoroughly addressed the reviewer comments and the manuscript is improved. It is an important addition to the literature.

7. PLOS authors have the option to publish the peer review history of their article (what does this mean?). If published, this will include your full peer review and any attached files.

Reviewer #1: Yes: Tim Gomersall

Reviewer #2: Yes: Joshua Grill

---

## [Editor Report · Acceptance letter]

13 Jan 2020

PONE-D-19-25712R1 

Clinicians’ communication with patients receiving a MCI diagnosis: the ABIDE project 

Dear Dr. Visser:

I am pleased to inform you that your manuscript has been deemed suitable for publication in PLOS ONE. Congratulations! Your manuscript is now with our production department. 

With kind regards,

on behalf of

Dr. Stephen D Ginsberg 

Section Editor

PLOS ONE